# Internal or External Word-of-Mouth (WOM), Why Do Patients Choose Doctors on Online Medical Services (OMSs) Single Platform in China?

**DOI:** 10.3390/ijerph192013293

**Published:** 2022-10-15

**Authors:** Jiang Shen, Bang An, Man Xu, Dan Gan, Ting Pan

**Affiliations:** 1College of Management and Economy, Tianjin University, Tianjin 300072, China; 2Business School, Nankai University, Tianjin 300071, China; 3School of Economics and Management, Hebei University of Technology, Tianjin 300071, China

**Keywords:** external word-of-mouth, online medical service, position ranking, optimal sequential search

## Abstract

(1) Background: Word-of-mouth (WOM) can influence patients’ choice of doctors in online medical services (OMSs). Previous studies have explored the relationship between internal WOM in online healthcare communities (OHCs) and patients’ choice of doctors. There is a lack of research on external WOM and position ranking in OMSs. (2) Methods: We develop an empirical model based on the data of 4435 doctors from a leading online healthcare community in China. We discuss the influence of internal and external WOM on patients’ choice of doctors in OMSs, exploring the interaction between internal and external WOM and the moderation of doctor position ranking. (3) Results: Both internal and external WOM had a positive impact on patients’ choice of doctors; there was a significant positive interaction between internal and third-party generated WOM, but the interaction between internal and relative-generated WOM, and the interaction between internal and doctor-generated WOM were both nonsignificant. The position ranking of doctors significantly enhanced the impact of internal WOM, whereas it weakened the impact of doctor recommendations on patients’ choice of doctors. (4) The results emphasize the importance of the research on external WOM in OMSs, and suggest that the moderation of internal WOM may be related to the credibility and accessibility of external WOM, and the impact of doctor position ranking can be explained by information search costs.

## 1. Introduction

Healthcare is necessary in people’s lives. Due to of the rapid increase in medical demand, the shortage and imbalance of medical resources have become urgent problems that need to be solved. These problems are particularly prominent in developing countries. According to the National Bureau of Statistics of China, in 2020, the number of urban health technicians per ten thousand people was 115, whereas the number of rural health technicians per ten thousand people was 52. A benefit of the rapid development of internet technology is the rapidly growing online healthcare communities (OHCs) that open up new possibilities for healthcare. OHCs are internet-based healthcare platforms, where doctors can provide patients with online medical services (OMSs) such as online health consultations, and share their health knowledge. At the same time, patients can remotely access these OMSs, learn about peers’ medical experiences, and receive emotional support [1]. Therefore, patients not only obtain OMSs through the OHCs, but also search for doctor information in the OHCs. According to this online information, patients gain access to internal WOM about doctors and can make an informed decision [2]. Compared with traditional healthcare, OMSs are unlimited in terms of time and space, which can help to alleviate the imbalance of medical resources [1,3]. Recently, the outbreak of COVID-19 highlighted the importance of OMSs, which reduced the need for patients to go to hospitals.

According to the optimal sequential theory, consumers search for information about products or services to evaluate the expected utility of the products or services and inform their decisions [4]. In OHCs, doctor information is divided into patient-generated information and system-generated information [5,6]. Patient-generated information provides information to patients about the services that other patients have received, and system-generated information includes basic information about the doctor, online activities, and comprehensive evaluation provided by OHCs. Researchers have suggested that seeing a doctor’s information in OHCs reflects the doctor’s trustworthiness [5], personal qualities [7], capital [8,9], service quality and attitude [6,8,10,11], reputation [7,12,13,14,15], and online efforts [10,13,16]. Patients can also learn about doctors’ interactions with patients [9,11,17], knowledge sharing [14,15,18,19], and log-in behavior [16,20]. This information generates a doctor’s internal WOM and has proven to have a positive impact on patient choices in OMSs [5,6,7,8,9,10,11,12,13,14,15,16,17,18,19,20]. In addition, some studies involving patient choices of doctor teams online discussed the impact of team diversity [21,22]. These studies have discussed the influence of internal WOM on patient choices of doctors in OMSs, whereas the impact of external WOM is usually ignored.

However, internal WOM is not the only way for patients to choose doctors in the OHCs. Patients can also access external WOM through relatives, doctors, third-party platforms, and traditional media to evaluate doctors. Then patients can directly purchase the medical services provided by the doctors without searching for information of other doctors. They are choosing according to external WOM. For example, when patients finish medical services offline, they can choose the same doctors in OHCs to obtain follow-up medical services [23]. Previous literature showed that the external WOM generated by third-party websites had a positive impact on consumer choice in the case of high-participation products [24] and low-participation products [25]. However, there is still a lack of evidence in the field of OMSs. Compared with products such as books and electronic devices, medical services involve human health, so patients will make more careful choices. Our objective was to survey the impact of internal and external WOM on patient choices of doctors in OMSs. Therefore, our first research question is:

(1) Does the external WOM impact patient choices of doctors in OMSs?

The interaction between internal and external WOM has also attracted the attention of researchers. Park et al. [26] suggested that a good external WOM generated by third-parties enhances the effect of internal WOM on consumer choices. The work of Chen et al. [27] demonstrated that the impact of consistency of user reviews on both internal and external platforms is positive. However, a study by Cao et al. reported conflicting results: the effects of online reviews on hotel revenue was weakened when the hotel had a good American Customer Satisfaction Index (ACSI) [28]. Another study discussed the interaction between third-party free sampling and internal WOM, and the interaction between third-party comments and internal WOM, indicating that the interaction may depend on whether consumers from different channels tended to give comments [29]. Although research has shown that online information has an impact on patients’ offline treatment decisions [30], research on internal and external WOM is still lacking in the case of OMSs. We hope to expand the research to OMSs, and further investigate the reasons for these different results. Our second research question is:

(2) What is the interaction between internal and external WOM on patient choices of doctors?

The impact of doctor position ranking has also largely been ignored by researchers in the field of OMSs. The effects of the ranking mean that consumers tend to prefer the products or services with top-ranking doctors [31]. These products or services are more visible to consumers and are considered to have higher quality [32,33]. However, some studies showed that the effects of position ranking were negatively related to the conversion rate [34,35,36]. The effects of position ranking on consumer choices were not significant in the cases of music and movies, where consumer preferences are dramatically different [37]. Due to increasing promotional and fake news being posted on consumer review websites, the position ranking is becoming increasingly biased and can be overestimated [32,38]. Thus, the impact of position ranking on consumer choices needs to be carefully considered. The influence of position ranking on patient information search and purchase behavior can be explained in two ways: signals [39,40] and information search cost [41]. Signals means that top-ranking doctors may mean higher evaluation and are thus, chosen more by consumers; the information search cost is the effort that consumers must pay to obtain and understand the information in the search process. Signals affect patients based on both internal and external WOM, whereas the information search cost only affects patients based on internal WOM. Thus, this paper discusses the moderating effect of position ranking in OMSs. Therefore, our third research question is:

(3) How does the position ranking of doctors moderate the impact of internal and external WOM on patient choices of doctors online?

To achieve the above-mentioned objectives, we conducted an empirical study based on the optimal sequence search theory. The rest of this paper is organized as follows: Section 2 presents the theoretical basis and hypotheses, Section 3 includes research methods, such as data sources, variables, and models, then the main results are discussed in Section 4. Finally, the conclusion summarizes the whole work of the study, including the main findings, theoretical and practical significance, and limitations.

## 2. Theoretical Basis and Hypotheses

This paper is based on the optimal sequential search theory from economics. The optimal sequential search theory is used to solve the problem of searching for the best outcome from alternative sources with different properties [42]. It assumes that people sample and evaluate in sequence and stop sampling when the expected marginal cost is higher than the marginal utility. After stopping sampling, people choose the object with the highest evaluation utility. This model has been widely used to describe the information search and product purchase behavior of online consumers and help provide the best product display list [4,43]. Prior studies have shown that this model can accurately predict actual product sales ranks and consumer demand [44,45], estimate consumer information and product purchase behavior [41,44,46,47], and explain the impact of internal WOM and position ranking on consumers’ choices online [31,38,48]. In addition, the theory establishes the relationship between outside options and internal search. As internal search and outside options are based on internal and external WOM, respectively, this theory helps us incorporate internal WOM, external WOM, and doctor position ranking into our research framework at the same time.

Based on the optimal sequential search theory, the process for patients to choose a certain doctor in OMSs can be described as follows: First, the patients decide whether to search up doctor information in the OHC, which is related to the expected benefit of an internal WOM search and benefits of outside options. When the benefits of outside options are not higher than the expected benefit of an internal WOM search, patients will search up doctor information. Second, once patients decide to start the search, they will search for doctor information in order. When patients think that the expected information search cost is higher than the expected utility, they will stop their search behavior [47]. Then, patients will retain the doctors with the highest internal search utility, unless the external options are more profitable, which highlights the positive relationship between the internal WOM and patients’ choice of doctors online. This view has been validated in prior empirical studies [9,13]. Thus, we hypothesize that:

**H1.** 
*Internal WOM has a positive impact on patient choices of doctors in OMSs.*


The outside options mean that patients choose doctors by means other than an internal search for information about doctors in one OHC, such as other OHCs, offline medical services, or a known doctor. When directly selecting a known doctor as an outside option, doctors’ external WOM determines the utility of this outside option. The higher utility of this outside option urges patients to directly select doctors instead of searching for other doctors in the OHC, which increases the probability of doctors being selected. Current studies also provide some evidence for this view. A retailer’s external WOM has been proven to play a more important role in the information search process [24,25]. Peer and expert advice can impact the information search behaviors of people [49]. Based on the above arguments, we propose that:

**H2.** 
*External WOM in OHCs is positively related to patients’ choice of online doctors.*


For high-involvement products, consumers tend to collect comprehensive information [21]. When patients log in to OHCs to choose a known doctor, they can also learn about the doctor’s internal WOM. Then, patients will update their expected utility [50]. When the patients reduce the expected utility of outside options due to bad signals from internal WOM, patients may change their decision, and turn to the internal search. A good internal WOM enhances consumer confidence in outside options. Prior studies have shown that the internal WOM of doctors has a positive impact on patient online follow-up intentions [23], and patient decisions on doctors recommended by their neighbors [51]. Thus, we propose that:

**H3.** 
*The interaction of internal and external WOM is positive.*


Due to the information search cost, consumers do not consider all the sets of products or services, but only the products or services at the top of the list [52]. Optimal sequential search also means that patients need to spend more time and pay greater cognitive costs before searching the information of doctors who are bottom-ranking. Searching for doctors with a bottom position ranking also exacerbates information overload and causes consumers to discontinue the online health information search [53]. Thus, doctors with top-ranking are more likely to be found by patients and are substantially related to high click-throughs. This result has been shown in a variety of contexts, such as yellow page ads, music choices of unknown songs, Google listings, movie or hotel listings, and so on [31,34,38,54,55]. Therefore, the external WOM of top-ranking doctors is more easily accessible to patients than doctors who are bottom-ranking. Thus, we propose that:

**H4.** 
*The position ranking of doctors online enhances the impact of internal WOM on patients’ choice of doctors in OMSs.*


If the doctor position ranking effect is interpreted as information search cost, the information search cost can change the importance of information sources to retailers: one source of information will become more important when it is difficult to obtain information from other sources. For example, the impact of online reviews is stronger in the case of less popular games, because it is difficult for consumers to obtain relevant information from alternative information sources [56]. Therefore, for doctors who are bottom-ranking, it is difficult for patients to obtain their internal WOM, which enhances the impact of external WOM. Thus, we propose that:

**H5a.** 
*The position ranking of doctors online weakens the impact of external WOM on patients’ choice of doctors in OMSs.*


The position ranking can also be considered an information cascade, and provides a positive signal [39,40]. Studies suggest that top-ranking positions will enhance the reliability of the results [57,58].

From this point of view, top-ranking positions can enhance the credibility of doctors’ external WOM. People are more likely to adopt information from reliable sources [59]. Thus, we propose that:

**H5b.** 
*The online position ranking of doctors positively moderates the relationship between doctor external WOM and patient choice of doctors.*


Figure 1 gives the model of our study.

## 3. Research and Methodology

### 3.1. Data Collection

Data collected from a leading OHC in China, www.haodf.com, was used to verify the hypothesis. This platform has gathered about 240,000 real-name registration of doctors from different hospitals and served more than 74 million patients by the end of 2021. We chose this platform as the research sample for the following reasons: (1) The platform is a popular OHC in China, and widely used in relevant research [8,9,11,17,18,60,61,62]; (2) The platform provides reasons why some health consumers choose a doctor in the patient review section; and (3) There are enough patients on the platform to vote for doctors. Figure 2 presents an example of one physician’s patient review section; this section includes the purpose of patients, treatment, efficacy satisfaction, attitude satisfaction, costs, state of illness, way of registration, and reasons for patient choice.

Figure 3 gives the selection of reasons of health consumers in the platform. According to Figure 3, there are five reasons for consumers to choose a doctor: online reviews, relatives’ recommendations, doctors’ recommendations, random registration, and others. Patients can choose at least one option as their reason after online medical consultation.

The sample included 4435 doctors specializing in the areas of cervical spondylosis, myopia, infertility, cerebral ischemic stroke, hypertension, and diabetes mellitus, which often afflict patients for a long time. These diseases are not acute, and it is feasible for patients to seek treatment online. Among them, cervical spondylosis, myopia and fertility are common in young people, and the last three diseases are common in the elderly. Therefore, it is representative sample. The data was collected from June to August 2021, and 28,386 patients voted during this period. All the data were publicly visible, and we desensitized the data in subsequent storage and use processes.

### 3.2. Variable Description

The patients who chose doctors due to “other” were manually divided into internal, external, and random based, according to the specific reasons provided by them. Then, the number of doctors’ patients in period *t* was used to evaluate the patients’ choice of doctors (*PatientsC*). A prior study suggested that patient votes are considered to be a reflection of doctors’ WOM [23]. Therefore, we used the number of patients based on internal WOM and patients who chose doctors based on online reviews to evaluate doctors’ internal WOM (*InWOM*). The number of patients who relied on external WOM and patients who chose doctors based on relatives’ and doctors’ recommendations were used to represent doctors’ external WOM (*ExWOM*). To further analyze the impact of external WOM, the external WOM was divided into relative-generated, doctor-generated, and third-party-generated WOM, expressed by *Corelatives*, *CoDoctors*, and *Others*, respectively. Position ranking (*Rank*) of doctors was the moderator variable; we used the derivative of the doctors’ real ranking provided by websites. In this way, a good ranking would have a higher value. Doctor titles (*Title*) were added to the model as control variables, and if a doctor’s title was chief physician, the value of a doctor’s title was 1; otherwise, it was 0. Other variables included the price of doctor services (*Price*), the number of articles published by doctors in OHCs in period *t* (*Article*), and the doctors’ comprehensive heat (*Heat)*. The doctors’ comprehensive heat means the degree of recommendation by patients from different disease fields in the OHC, which was similar to rating. Table 1 gives a summary of the variables.

### 3.3. Estimation Model

Seven models were hierarchically developed to verify our hypothesis. Model (1) including only independent variables was used to test H1 and H2. Model (2) was used to test H3, and model (3) included all the control variables based on model (2). Compared with model (3), the external WOM was replaced by the number of patients who chose doctors based on relative and doctor recommendations and other information channels in model (4), to discuss the relationship between doctors’ internal WOM and external WOM. Model (5) was tested to verify the moderating effect of position rankings (H4, H5a, and H5b), and model (6) included all the control variables based on model (5). Lastly, in model (7), based on model (6), the external WOM was replaced by the number of patients who chose doctors based on relative and doctor recommendations and other information channels. The models are as follows:(1)$$PatientsC_{it}=\beta_{0}+\beta_{1}InWOM_{it}+\beta_{2}ExWOM_{it}+\beta Z_{it}$$
(2)$$PatientsC_{it}=\beta_{0}+\beta_{1}InWOM_{it}+\beta_{2}ExWOM_{it}+\beta_{3}InWOM_{it}\times ExWOM_{it}+\beta Z_{it}$$
(3)$$\begin{array}{c} PatientsC_{it}=\beta_{0}+\beta_{1}InWOM_{it}+\beta_{2}ExWOM_{it}+\beta_{3}Rank_{it}+\beta_{4}Title+\beta_{5}Price+\beta_{6}Article_{it-1}\\ +\beta_{7}Heat_{it-1}+\beta_{8}InWOM_{it}\times ExWOM_{it}+\beta Z_{it} \end{array}$$
(4)$$\begin{array}{c} PatientsC_{it}=\beta_{0}+\beta_{1}InWOM_{it}+\beta_{2}Corelatives_{it}+\beta_{3}CoDoctors_{it}+\beta_{4}Others_{it}+\beta_{5}Rank_{it-1}+\\ \beta_{6}Title+\beta_{7}Price+\beta_{8}Article_{it-1}+\beta_{9}Heat_{it-1}+\beta_{10}Rank_{it-1}\times InWOM_{it}+\beta_{11}InWOM_{it}\times \\ CoRelatives_{it}+\beta_{12}InWOM_{it}\times CoDoctors_{it}+\beta_{13}InWOM_{it}\times Others_{it}+\beta Z_{it} \end{array}$$
(5)$$\begin{array}{c} PatientsC_{it}=\beta_{0}+\beta_{1}InWOM_{it}+\beta_{2}ExWOM_{it}+\beta_{3}Rank_{it-1}+\beta_{4}Rank_{it-1}*InWOM_{it}+\\ \beta_{5}Rank_{it-1}\times ExWOM_{it}+\beta Z_{it} \end{array}$$
(6)$$\begin{array}{c} PatientsC_{it}=\beta_{0}+\beta_{1}InWOM_{it}+\beta_{2}ExWOM_{it}+\beta_{3}Rank_{it-1}+\beta_{4}\mathrm{Title}+\beta_{5}\mathrm{Price}+\beta_{6}{\mathrm{Article}}_{it}+\\ \beta_{7}{\mathrm{Heat}}_{it-1}+\beta_{8}{\mathrm{Rank}}_{it-1}\times InWOM_{it}+\beta_{9}{\mathrm{Rank}}_{it-1}\times ExWOM_{it}+\beta Z_{it} \end{array}$$
(7)$$\begin{array}{c} PatientsC_{it}=\beta_{0}+\beta_{1}InWOM_{it}+\beta_{2}Corelatives_{it}+\beta_{3}CoDoctors_{it}+\beta_{4}Others_{t}+\beta_{5}Rank_{it-1}+\beta_{6}Title+\\ \beta_{7}Price+\beta_{8}Article_{it}+\beta_{9}Heat_{it-1}+\beta_{10}Rank_{it-1}\times InWOM_{it}+\beta_{11}Rank_{it-1}\times CoRelatives_{it}+\\ \beta_{12}Rank_{it-1}\times {\mathrm{CoDoctors}}_{it}+\beta_{13}Rank_{it-1}\times {\mathrm{Others}}_{it-1}+\beta Z_{it} \end{array}$$
where *β* parameters are the coefficients to be estimated and Z are the vector-controlling variables. In the models, period *t* was set to two weeks. In this way, a sufficient number of new patients, WOMs, and articles could be generated in period *t*, whereas doctors’ position rankings and comprehensive heat would not significantly change. We used the fixed-effect model (FEM) method to test the models. The fixed-effect model uses within-individual variation to identify coefficient estimates, and was used to control potential unobserved physician-level heterogeneities, mitigate certain endogeneity, and rule out spurious relationships [63]. To further test the robustness of our findings, we also performed hierarchical regression using ordinary least squares (OLS) [62]. Before the FEM and OLS, the data was normalized.

## 4. Results

### 4.1. Hypotheses Testing

Table 2 gives the hierarchical regression results using FEM.

According to the results of model (1), internal WOM (*InWOM*, *β* = 0.263, t = 22.649, *p* < 0.001) and external WOM (*ExWOM*, *β* = 0.269, t = 22.340, *p* < 0.001) were positively related to patient choice of doctors, and the impact was significant. The above results support H1 and H2. Both internal and external WOM had a positive impact on patient choices of doctors.

Model (2) was used to assess the interaction between internal and external WOM. The results suggested that the interaction effect between internal and external WOM (*InWOM* × *ExWOM*, *β* = −0.002, t = −1.450, *p* = 0.147) was negative, but not significant, which did not support our H3. According to the results of model (4), the interaction between internal WOM and other information channels (*InWOM* × *Others*, *β* = 0.010, t = 3.619, *p* < 0.001) was significant and positive; however, the results of relative recommendations (*InWOM* × *CoRelatives*, *β* = 0.000, t = 0.063, *p* = 0.949) and doctor recommendations (*InWOM* × *CoDoctors*, *β* = −0.002, t = −0.487, *p* = 0.626) were not significant.

The moderating role of position rankings was tested based on model (5). We found that the moderating effect of position ranking on the relationship between internal WOM (*Rank* × *InWOM*, *β* = 0.029, t = 4.488, *p* < 0.001) and patient choices of doctors was significantly positive. This result supported H3; the position rankings of doctors positively moderated the relationship between the internal WOM and patient choices of doctors. On the other hand, the interaction terms between position ranking and external WOM (*Rank* × *ExWOM*, *β* = −0.008, t = −3.090, *p* = 0.003) were negative and significant, which supported H4a. The results of model (7) suggested the moderating effect of position ranking on the relationship between doctor recommendations (*Rank* × *CoDoctors*, *β* = −0.013 t = −3.295, *p* = 0.001) and patient choice was significantly negative. On the other hand, the interaction terms between position ranking and relative recommendations (*Rank* × *CoRelatives*, *β* = 0.002, t = 0.306, *p* = 0.759) and the interaction terms between position ranking and other information channels (*Rank* × *Others*, *β* = 0.005, t = 1.389, *p* = 0.165) were insignificant.

Model (3) and model (6) presented that, after adding the control variables, the results were generally consistent with the models without control variables, which showed that the model was robust.

### 4.2. Robustness Checks

Table 3 shows the results of hierarchical regression using OLS. We found that most of the effects we studied were quantitatively consistent with our main findings. Therefore, our results were credible and robust.

## 5. Discussion

### 5.1. Key Findings and Theoretical Implications

This study explored the impact of internal and external WOM on patient choices of doctors in OMSs through H1 and H2. The results suggested that both internal and external WOM had a positive impact on patient choices of doctors, supporting our H1 and H2. Studies have shown that internal WOM has a positive impact on patient choices of doctors in OMSs [5,18], whereas our study emphasized the importance of external WOM in OMSs. Prior studies have shown the positive impact of external WOM on consumer choices in the case of digital cameras [24] and books [25]. Our study provides new evidence in the field of OMSs. Besides third-party-generated WOM, this paper also suggests that relative-generated and doctor-generated WOM can have a positive impact on patient choices of doctors. This is significant as relative-generated and doctor-generated WOM are private and offline. Therefore, we also contribute to the optimal sequential search theory. The optimal sequential search theory has been used to predict sales [44,55]. In these search models, outside options influenced by the external WOM are ignored. Our research suggests that such outside options may lead to bias in the estimation of the sales of products or service selections.

Second, H3 discussed the interaction effect between internal WOM and external WOM. Our results suggested that the interaction between internal and external WOM as not significant. Further analysis showed that internal WOM enhanced the impact of third-party generated WOM on patient choices of doctors in OMSs, which is consistent with the research from the Centre National d’Études des Télécommunications (CNET) [26,27,60], and contrary to a study from ACSI [28]. In the process of continuously searching for information about products or services, consumers update their expected utility for products or services at each search occasion; when consumers believe that the existing information is accurate enough or the further information search cost is too high, they will stop searching [50]. In medical services, patients may have more trust in doctors and their relatives than in thirty-parties [64], and the privacy of medical services will lead to limited information about patients’ medical experiences in internal WOM [65]. This weakens the impact of internal WOM on doctor-generated and relative-generated WOM. When sufficiently credible external WOM is available to all the consumers, external WOM and internal WOM can replace each other, just as in the case of ACSI [28]. Therefore, the interaction between internal and external WOM is related to the credibility and accessibility of external WOM. In addition, the different results of the relationship between external and internal WOM suggest that it is necessary to learn which online information may be related to outside options and what the relationship between them is. This helps to explore the role of outside options in the optimal sequential search theory.

Lastly, the moderating effect of the position ranking of doctors in the OHCs was explored through H4 and H5. The results supported our H4 and H5a, which suggest that doctor position ranking affects patient information search behavior by changing the information search cost. The finding that doctor position ranking significantly enhanced the impact of internal WOM on patient choices of doctors was consistent with the results of a study about software [40], but contrary to another study on e-books [25]. Prior research showed that the impact of ranking was weakened when consumer preferences for products were significantly different [37]. Consumers need to browse as many products as possible to compare products with high homogeneity. The optimal sequential search theory showed that, under the same conditions, the greater the variance of alternative returns, the fewer options that are searched. In online medicine, the services provided by doctors are nearly homogeneous. Further analysis of H5 showed that position ranking only significantly weakened the impact of doctor recommendations on patient choices of doctors, and the moderating effects were not significant between the other two external WOM (relative recommendations and others) and patient choices of doctors. It may be because doctors with top online ratings or rankings will attract more patients [60]. Therefore, compared with doctors with bottom-ranking positions, top-ranking doctors are busier. In addition, specific reasons given by patients in our study showed that an important reason why doctors recommend other doctors to patients is that they do not have time.

### 5.2. Practical Implications and Suggestions

This study has some important implications for the provider of the OHCs. First, the results show that position ranking has a positive effect on the relationship between internal WOM and patient choices of doctors, which means that the information search costs are significant [32]. Second, the interaction between internal and doctor-generated WOM and the interaction between internal and relative-generated WOM are non-significant, which suggests that when patients directly search for a known doctor in the OHCs, the internal information may lack reference value for patients. Lastly, top-ranking doctors occupy the main online resources, whereas doctors with bottom-ranking positions benefit more from external WOM than from OHCs, which leads to a vicious circle [60]. A good doctor may not be able to stand out because of the late opening of online services, because it may be hard for them to get a top ranking in the first place. In turn, this may cause doctors to lose enthusiasm to continue to use the online platform. As doctors’ energies are limited, too many patients may reduce the communication efficiency between doctors and patients.

Therefore, OHCs must reduce information search costs and improve the availability of information. Past research has suggested that providing patients with useful information promotes patients’ health information retrieval behavior [66]. Improving the comprehensibility of the information may be a good idea. Information about online doctors’ service delivery processes, such as the doctor’s response speed, interaction frequency, and interaction depth has proven to help patients decide in OMSs [1,67]. However, understanding this information from the huge amounts of internal information available is undoubtedly a challenge for patients. OHCs can easily extract the information that patients are interested in and provide it to patients, thus, increasing the possibility for patients to make decisions using internal information. Second, OHCs can also reduce information search costs by making doctor position ranking easier. For example, OHCs can explain to the patients the primary basis for position ranking. When consumers are unaware of such ranking rules, information search costs will increase [68]. This also helps to increase the feasibility that doctors with bottom-ranking positions will benefit from OHCs and patients’ trust in the position ranking. Lastly, a diversified recommendation mechanism is necessary for the OHCs, such as recommendation system. A recommendation system could be considered a supplement to patients’ outside options. This could help patients choose more appropriate doctors (not only in service quality but also in the available service time of doctors).

We also have some implications for doctors and patients. For doctors, improving their internal and external WOM both help them achieve higher performance in OMSs. The position ranking of doctors in OHCs is important. If doctors cannot be in a favorable position in an OHC, their information is difficult to be noticed by other patients in OHCs. Thus, it is wise for doctors to consider the OHC as a new communication channel with their patients or choose to develop in other OHCs. For patients, the most popular doctor may not always be the most suitable doctor, so it may be beneficial to try to search for more doctors in OHCs.

### 5.3. Limitations and Future Directions

Although this research has certain theoretical and practical implications, there are some limitations. First, the generalizability of the results is limited. The results are based on data from six diseases and data was only collected from one online platform, haodf.com. Although the selected diseases and platforms are considered to be representative of China, further analysis of more disease areas and multiple platforms is needed in the future. In addition, research on different national backgrounds is also necessary. For example, in the UK, patients are first seen by a primary care physician who would refer them to a secondary care specialist. Different methods of choosing doctors may lead to different results. Second, although the results suggest the importance of external WOM in OMSs, the relationship between internal and external WOM, position ranking, and decision-making needs more in-depth research. In the future, we hope to describe patient search and decision-making in more detail, to establish a predictive relationship between external WOM and patient decisions. Third, not all patients vote for doctors, so further analysis of the characteristics of patients who vote is needed. Due to the importance of doctors in providing OMSs, further research should provide long-term observation and analysis of doctors’ online performance to study the behavior of doctors in online medical activities. Other influencing factors may also need to be discussed in future research, such as the doctor’s age, region, etc.

## 6. Conclusions

This study explored the impact of internal and external WOM on patient choices of doctors in OMSs and discussed the moderating effect of position rankings by developing a theoretical model. The data from a mainstream online medical platform in China was used to test the model. According to the results, several key conclusions were made. First, external WOM played an important role in OMSs. Second, internal WOM enhanced the impact of third-party-generated WOM, whereas it had no significant effect on the impact of WOM generated by relatives and doctors. The results showed that the moderation of internal WOM may be related to the credibility and accessibility of external WOM. Lastly, the position rankings moderated the relationship between WOM and patient choices in OMSs. Top-ranking positions enhanced the impact of internal WOM on patients’ choices, weakened the impact of doctor-generated WOM, and had no significant effect on the impact of external WOM generated by relatives and third-parties. This means that doctor position ranking affects patient information search behavior through the information search cost. The benefits of doctors at the bottom of the ranking list from online platforms were limited.

## Figures and Tables

**Figure 1 ijerph-19-13293-f001:**
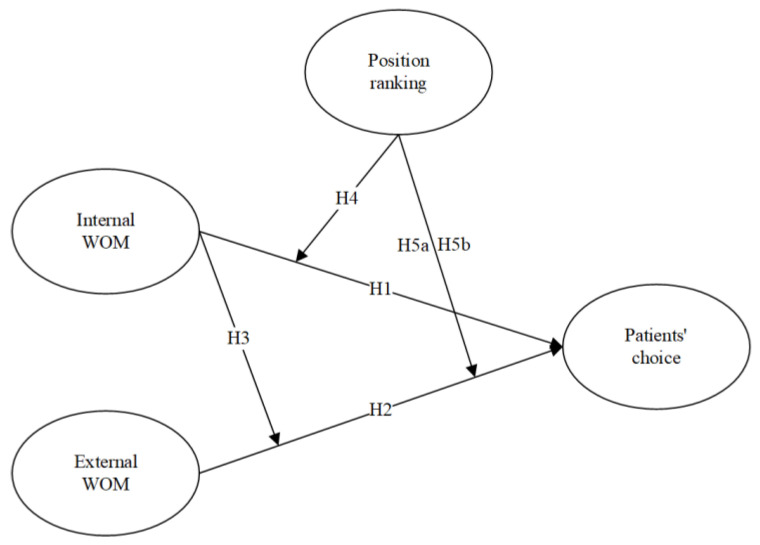
Research model of this study.

**Figure 2 ijerph-19-13293-f002:**
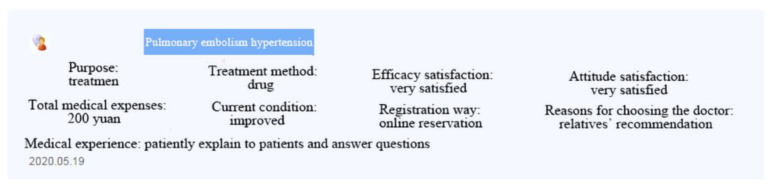
Doctor *patient review* section.

**Figure 3 ijerph-19-13293-f003:**
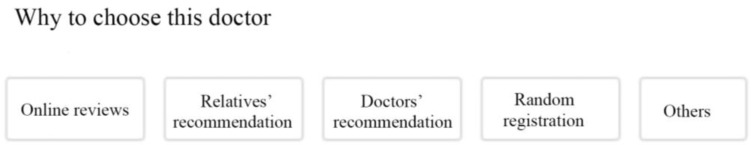
Selection reasons.

**Table 1 ijerph-19-13293-t001:** Variable description.

Variables		Description
Dependent variable	*PatientsC*	The number of doctor’s patients in period *t*.
Independent variable	*InWOM*	The number of patients who chose doctors based on internal WOM in period *t*.
	*ExWOM*	The number of patients who chose doctors based on external WOM in period *t*.
	*CoRelatives*	The number of patients who chose doctors based on relative recommendations in period *t*.
	*CoDoctors*	The number of patients who chose doctors based on doctor recommendations in period *t*.
	*Others*	The number of patients who chose doctors based on other information channels in period *t*.
Control variable	*Title*	Is the doctor a chief physician?
	*Price*	The price of doctor’s services
	*Article*	The number of articles published by doctors in OHCs in period *t*.
	*Heat*	Doctor’s comprehensive heat.
Moderator variable	*Rank*	The derivative of the doctor’s real ranking provided by websites.

**Table 2 ijerph-19-13293-t002:** Hierarchical regression results using FEM.

	Model (1)	Model (2)	Model (3)	Model (4)	Model (5)	Model (6)	Model (7)
*InWOM*	0.275 ***	0.288 ***	0.273 ***	0.261 ***	0.267 ***	0.263 ***	0.271 ***
*ExWOM*	0.284 ***	0.287 ***	0.266 ***		0.284 ***	0.269 ***	
*CoRelatives*				0.145 ***			0.141 ***
*CoDoctors*				0.055 ***			0.059 ***
*Others*				0.129 ***			0.145 ***
*Rank* × *InWOM*					0.029 ***	0.028 ***	0.021 ***
*Rank* × *ExWOM*					−0.008 **	−0.010 **	
*Rank* × *CoRelatives*							0.002
*Rank* × *CoDoctors*							−0.013 ***
*Rank* × *Others*							0.005
*InWOM* × *ExWOM*		−0.002	−0.000				
*InWOM* × *CoRelatives*				0.000			
*InWOM* × *CoDoctors*				−0.002			
*InWOM* × *Others*				0.010 ***			
*Rank*			0.005	0.001	−0.003	−0.001	−0.019
*Title*			−0.002	0.000		−0.003	−0.002
*Price*			0.017	0.029		0.020	0.025
*Article*			0.087 ***	0.087 ***		0.087 ***	0.088 ***
*Heat*			0.292***	0.293***		0.290 ***	0.314 ***
R-Square	0.213 ***	0.213 ***	0.209 ***	0.214 ***	0.213 ***	0.211 ***	0.216 ***

Note: ** *p* < 0.01. *** *p* < 0.001. There are 4435 observations.

**Table 3 ijerph-19-13293-t003:** Hierarchical regression results using OLS.

	Model (1)	Model (2)	Model (3)	Model (4)	Model (5)	Model (6)	Model (7)
*InWOM*	0.296 ***	0.347 ***	0.353 ***	0.345 ***	0.287 ***	0.291 ***	0.302 ***
*ExWOM*	0.348 ***	0.362 ***	0.367 ***		0.351 ***	0.361 ***	
*CoRelatives*				0.197 ***			0.183 ***
*CoDoctors*				0.085 ***			0.086 ***
*Other*				0.183 ***			0.192 ***
*Rank* × *InWOM*					0.033 ***	0.032 ***	0.025
*Rank* × *ExWOM*					−0.011 ***	−0.014 ***	
*Rank* × *CoRelatives*							0.001
*Rank* × *CoDoctors*							−0.016 ***
*Rank* × *Other*							0.003
*InWOM* × *ExWOM*		−0.008 ***	−0.009 ***				
*InWOM* × *CoRelatives*				−0.003			
*InWOM* × *CoDoctors*				−0.005			
*InWOM* × *Other*				0.003			
*Rank*			−0.001	−0.007	−0.002	−0.005	−0.023
*Title*			−0.082	−0.081		−0.078	−0.076
*Price*			0.019	0.032		0.024	0.032
*Article*			0.099 ***	0.099 ***		0.099 ***	0.100 ***
*Heat*			0.620 ***	0.633***		0.633 ***	0.658 ***
*R-Square*	0.266 ***	0.267 ***	0.287 ***	0.292 ***	0.268 ***	0.288 ***	0.294 ***

Note: *** *p* < 0.001. There are 4435 observations.

## Data Availability

Not applicable.

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
