# Peer review of "Internal or External Word-of-Mouth (WOM), Why Do Patients Choose Doctors on Online Medical Services (OMSs) Single Platform in China?"

_ijerph, 2022, doi:10.3390/ijerph192013293_

Round 1

Reviewer 1 Report

The article investigates the interplay of online and offline information (internal vs. external word-of-mouth) for decision making, using the example of patients' choice for a certain doctor within a Chinese Online Health Community (OHC). The article describe, that it is the first one to take the offline information into account, additionally to online rankings. By means of an empirically developed hierarchiacal model, five hypothesis were tested. Results show that both online and offline word of mouth play a role for patients' choice for a doctor. Several influencing factors are considered.

The introduction is devided into three parts. It is suggested to integrate points 1, 2 and 3 into one subheading "Introduction" to avoid repetition and to facilitate a stringent story line. Particularly the concept of OHCs and the relevance of the word-of-mouth may be made more explicit to the international readers. This might also enhance the understanding of the relevance of the invesigation and the reasearch questions. Literature about decision making and choice for a certain physician (online and offline) may be added. Furthermore, a many hypothesis were tested. It might be considered to focus the article on two or three aspects.

As a basic assumption, the article compares the voting of doctors to the voting of products and services. It should be described in more detail, how and if this comparison may be made in terms of ethical and methodological considerations. Treatment by a doctor can be considered as related to intimate private and health issues and may have a high impact on physical and psychological states, wheres the decision to buy e.g. a certain book may be considered as less critical for people to decide on. 

Regarding the methods, a single online platform is analyzed. A control group or a description of the special customers of this platform are recommended. As well, the decision for certain diagnosis may be justified, taking into account data protection issues of personal medical diagnosis. The presentation of the hierarchical models is clear, however not easy to interpret due to the high number of variables. As suggested before, a focus on certain aspects of the study might be considered.

The discussion focuses on the key findings and their pracitcal and theoretical implications.The reference to a higher order frame, such as decision making should be considered and would be in line with suggestions regarding the articles'  introduction. Limitations were descirbed in a rather short manner and might be extended with more detail and/ or literature. Generalizablility across medical systems of different countries is considered an important addition to the discussion. 

Author Response

Tanks very much for your evaluation of our review and your valuable comments. For more information, please see the attachment.

Reviewer 2 Report

The big problem with choosing doctors on the internet is that no one can see the brass plaque on the door advertising their qualifications: therefore, knowing that someone is good (or not) does tend to rely on recommendations, which can also be true false and positive or negative. Thus, word of mouth is a powerful way to influence people in their choices, and it may be for positive or negative reasons…

The problem of doctor availability is recognised to be a problem. In the UK, we have 230 doctors / 100,000 population [not broken down in to urban / rural], so the 115 or 52 in China is significantly lower. What are the population proportions? If 80% of Chinese population is considered urban, the medical deficit would be lower than if the rural proportion were 80%.

How doctors are selected will vary between countries. In the UK, patients are first seen by a primary care physician who would refer them to a secondary care specialist. It appears from the text that in China patients may directly self-refer to specialists. Some explanation of the context of choosing doctors in China might be useful to add to the manuscript. The use of online health communities may also vary by the age of the doctor. Younger more internet aware doctors may be more likely to use online health communities [possibly as part of an advertising campaign], than older established physicians [who do not need to advertise because they have already developed an external WOM reputation.

The difficulty of assessing the importance of external WOM is easy to understand: Things that are easy to measure often are measured [whether there are relevant or not], but things that are difficult to measure often are not, even when they are important: internal WOM will be easy to trace as it is online, external WOM will be difficult because it is not. Furthermore, external WOM has the potential to be more negatively powerful that the online ‘internal’ WOM because particularly negative information is less likely to be placed online for fear of harming the care of relatives already under care of a particular practitioner if a complaint can be linked to them, and because of fears of complaints of defamation. Consequently, online information would be likely to be more positive than perhaps it should be.

Ranking of doctors can also be a problem: How is the rank calculated. The best surgeon may in fact have poorer statistics because he is referred the most difficult cases with the lowest chance of survival, but he may get more of those patients to live even though his overall performance against those who only get the easy patients looks poor because those patients were less likely to die. Another confounder may be cost: patient’s choices may also be constrained by what they can afford, so some of the best doctors may get fewer patients because they charge more than some of the market can afford: again, this may be affected by the health economy in which they work so some context of the way that the Chinese market operates may be useful – for example are there price controls, or are doctors allowed unconstrained pricing flexibility

Thera re thus many extra factors that may need to be considered in a true to life model. The model shown in figure 1 is however appears to be an adequate start. I am not sufficiently skilled to comment on the statistical results shown – but they do appear to show some statistically significant findings.

Overall, this is an interesting study that clearly shows that many factors affect consumer choice and that some things that professionals might consider highly important are discounted by patients.

Author Response

Tanks very much for your evaluation of our review and your valuable comments. And your comments will also help us in future research.

As you mentioned: “How doctors are selected will vary between countries. In the UK, patients are first seen by a primary care physician who would refer them to a secondary care specialist. It appears from the text that in China patients may directly self-refer to specialists.” We believe that it is a valuable study to analyze and compare patients' information search and decision-making behaviors in different countries. And this may be helpful to the research of policies. “Younger more internet aware doctors may be more likely to use online health communities [possibly as part of an advertising campaign], than older established physicians [who do not need to advertise because they have already developed an external WOM reputation.” We very much agree with this view. Besides the age of the doctor, the region of the doctor may also have an impact. We hope to study these problems in the future. Then we describe the above contents in the part of limitation and future direction.

“The difficulty of assessing the importance of external WOM is easy to understand: Things that are easy to measure often are measured [whether there are relevant or not], but things that are difficult to measure often are not, even when they are important: internal WOM will be easy to trace as it is online, external WOM will be difficult because it is not. Furthermore, external WOM has the potential to be more negatively powerful that the online ‘internal’ WOM because particularly negative information is less likely to be placed online for fear of harming the care of relatives already under care of a particular practitioner if a complaint can be linked to them, and because of fears of complaints of defamation. Consequently, online information would be likely to be more positive than perhaps it should be.” The measurement of external reputation depends on the active search of the website and the cooperation of patients. And, we try our best to use an equivalent way to measure internal and external WOM. The measurement of internal and external word of mouth is based on their scope of influence in this paper.

“Ranking of doctors can also be a problem: How is the rank calculated. The best surgeon may in fact have poorer statistics because he is referred the most difficult cases with the lowest chance of survival, but he may get more of those patients to live even though his overall performance against those who only get the easy patients looks poor because those patients were less likely to die. Another confounder may be cost: patient’s choices may also be constrained by what they can afford, so some of the best doctors may get fewer patients because they charge more than some of the market can afford: again, this may be affected by the health economy in which they work so some context of the way that the Chinese market operates may be useful – for example are there price controls, or are doctors allowed unconstrained pricing flexibility.” The website did not give a clear explanation on how to rank doctors, although they indicated that it was affected by patient recommendations. While the doctors with the highest ranking will be in a more easily found position on the website, this is exactly what we need for research. For the price problem, we use the price variable as the control variable in the model, to exclude the influence of price. And in China, doctors are generally not allowed to set too high a price for consultation.

And combined with comments from other reviews, we have made the following changes to the article. We integrated the content of literature review into the introduction, and reorganized the introduction. The order of raising research questions is changed from 1) the impact of internal and external WOM, epically external WOM; 2) the moderation effect of position ranking; and 3) the interaction between internal and external WOM, to 1) the impact of internal and external WOM, epically external WOM; 2) the interaction between internal and external WOM; and 3) the moderation effect of position ranking, to make it more logical. Therefore, we also modified the narrative order of the following parts. In the part of theoretical background and hypotheses, we have strengthened the connection between theory and research model. We have integrated the key findings and theoretical contributions to enhance the interpretation of the results and highlight the research focus. And we added suggestions on OHC management decisions in the part of practical implications. We have also enriched the content of limitations and future direction. Finally, we have adjusted some expressions in the article, mainly about abbreviations, and corrected some errors or wrong expressions.

Thank you again for your valuable comments on our research.

Reviewer 3 Report

Please check the attached file

Author Response

(The authors gave the same response as above.)

Round 2

Reviewer 1 Report

Thank you for the revision of your manuscript. The paper has profoundly improved and the suggestions of the first review have been mainly met.

The introduction has benefited a lot from the revision. The understanding of WOM and OHCs is much better now for readers who are not familiar with the Chinese OHC-system.  Also the understanding of the research questions and hypotheses is much clearer. It is suggested to add a little context information about the "optimal sequence search theory" in the introduction - only in the discussion it is described, that it stems from the "sales context". It might be considered to shorten (and focus) the introduction and hypothesis-sections.

The methods section may be improved by including the complete description of analysis. In the results section, there were presented two versions of the models, without prior description of FEM and OLS in the methods section. This should be included there and justified, why both are needed and what they mean/ how they are interpreted/ which differences there are.

In the discussion section, the key findings should be grouped according to the hypotheses, not the tested models. Also, some more references might be included, when discussing possible implications. As well, a shortening of the discussion may be considered by the authors.

In total, I generally suggest a sound English review. Particularly I would like to point out to remove the frequently used "And" at beginning of a sentence. 

Author Response

(The authors gave the same response as above.)

Reviewer 3 Report

Good Job!

Author Response

Thank you for your recognition of our work. And according to other reviewers, we made some modifications as follows:

We reorganized the introduction and mentioned the content of “the optimal sequential search theory”. In the methods section, we advance the description of the model and reorganized it. At the same time, we added the prior description of FEM and OLS in this section. We reorganized the discussion according to the hypotheses, and more references were included. Finally, we refined the introduction, theoretical basis and assumptions, and key findings, as well as conducted a sound English review of the article.

We wish good health to you, your family, and your community.